# Quantifying the Cost of Reliable Photo Authentication via High-Performance Learned Lossy Representations

**Paweł Korus**[1,2] **& Nasir Memon**[1]
[1]Tandon School of Engineering, New York University, USA
[2]Department of Telecommunications, AGH University of Science and Technology, Poland
`{pkorus,memon}@nyu.edu`

## Abstract

Detection of photo manipulation relies on subtle statistical traces, notoriously removed by aggressive lossy compression employed online. We demonstrate that end-to-end modeling of complex photo dissemination channels allows for codec optimization with explicit provenance objectives. We design a lightweight trainable lossy image codec, that delivers good rate-distortion performance, comparable with the popular hand-crafted BPG, and has low computational footprint on modern GPU-enabled platforms. Our results show that significant improvements in manipulation detection accuracy are possible at fractional costs in bandwidth/storage. Our codec improved the accuracy from 37% to 86% even at very low bit-rates, well below the practicality of JPEG (QF 20).

## 1 Introduction

Increasing adoption of machine learning in computer graphics has rapidly decreased the time-frame and skill set needed for convincing photo manipulation. Point-and-click solutions are readily available for plausible object insertion (Portenier et al., 2019), removal (Xiong et al., 2019), sky replacement (Tsai et al., 2016), face editing (Portenier et al., 2018) and many other popular operations. While often performed with humorous or artistic intent, they can wreak havoc by altering medical records (Mirsky et al., 2019), concealing scientific misconduct (Gilbert, 2009; Bik et al., 2016; Bucci, 2018) or even interfering with democratic elections (Chesney & Citron, 2019).

Reasoning about photo integrity and origin relies on subtle statistical traces, e.g., fingerprints of imaging sensors (Chen et al., 2008), color interpolation artifacts (Popescu & Farid, 2005), or pixel co-occurrence patterns (Marra et al., 2019b; Mayer & Stamm, 2019). Unfortunately, such traces are commonly destroyed during online dissemination, since social networks are forced to aggressively compress digital media to optimize storage and bandwidth expenditures - especially on mobile devices (Cabral & Kandrot, 2015). As a result, detection of photo manipulations online is notoriously unreliable. Some platforms perform forensic photo analysis at the ingress (Truepic, 2019), but it may already be too late. Existing photo compression standards, like JPEG, optimize for human perception alone and aggressively remove weak micro-signals already at the device.

We demonstrate that huge gains in photo manipulation detection accuracy are possible at low cost by carefully optimizing lossy compression. Thanks to explicit optimization, fractional increase in bitrate is sufficient to significantly increase the detection accuracy. We build upon the work of Korus & Memon (2019) and use their toolbox for end-to-end modeling of photo dissemination channels. We design a lightweight and high-performance lossy image codec, and optimize for reliable manipulation detection - a backbone of modern forensic analysis (Wu et al., 2019; Mayer & Stamm, 2019). Interestingly, the model learns complex frequency attenuation patterns as simple inclusion of high-frequency information turns out to be insufficient. This suggests new directions in ongoing efforts to revisit the standard rate-distortion paradigm (Blau & Michaeli, 2019).

We believe such solutions could be useful for social media platforms, photo attestation services, or insurance companies, which may exploit asymmetric compression setups and acquire photos from smart-phones in analysis-friendly formats. In terms of rate-distortion, our model is comparable

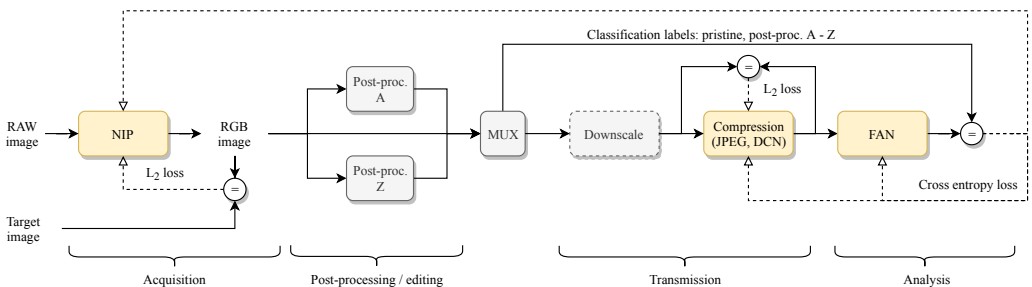

Figure 1: A generic end-to-end trainable model of photo acquisition and dissemination: camera ISP is modeled by a neural imaging pipeline (NIP); manipulation detection is performed by a forensic analysis network (FAN); the channel may use either JPEG or a trainable deep compression network (DCN). Potentially trainable elements are shown in yellow.

with modern hand-engineered codecs, like BPG (Bellard, 2014) which delivers only slightly better results. On GPU-enabled platforms, our codec can be faster, even without low-level optimization.

## 2 RELATED WORK

**Learned Compression:** Rapid progress in deep learning has rekindled interest in lossy image compression. While some studies consider fully end-to-end solutions dispensing with conventional entropy coding (Toderici et al., 2017), the most successful solutions tend to be variations of auto-encoders combined with context-adaptive arithmetic coding. Such codecs have recently surpassed state-of-the-art hand-crafted solutions (Rippel & Bourdev, 2017; Mentzer et al., 2018). Adoption of generative models allows to hallucinate unimportant details, and reach extreme compression rates while maintaining good perceptual quality (Agustsson et al., 2018). This research direction makes explicit provenance objectives increasingly pressing.

**Compression vs High-level Vision:** JPEG compression is commonly used for data augmentation to retain high machine vision performance on compressed images. Despite this, severe compression is known to degrade accuracy (Dodge & Karam, 2016), and restoration techniques are often used to mitigate the problem (Wang et al., 2016). Some studies optimize JPEG compression to encode semantically salient regions with better quality in a format-compliant way (Prakash et al., 2017). Researchers also explore trainable variations of the JPEG codec optimized for minimal performance degradation and low power use in IoT devices (Liu et al., 2018). In high-volume applications, computational footprint can be reduced by running high-level vision directly on the DCT coefficients (Gueguen et al., 2018). Adoption of trainable latent representations gives more flexibility and allows for end-to-end training (Torfason et al., 2018).

**Optimization of Photo Dissemination Channels:** Large volume of photos shared online spawned the need to aggressively optimize all steps of photo dissemination (uplink, downlink and storage). Social media platforms already rely on in-house solutions (Facebook, 2018), and employ extreme measures, like header transplantation, to minimize overhead and improve user experience (Cabral & Kandrot, 2015). The platforms actively engage in research and development of image compression, including optimization of the standard JPEG codec (Google, 2016), development of new backward-compatible standards like JPEG-XL (Rhatushnyak et al., 2019), and development of entirely new codecs - both hand-engineered (e.g., WebP) and end-to-end trained (Toderici et al., 2017).

## 3 END-TO-END TRAINABLE PHOTO DISSEMINATION MODEL

We build upon a recently published end-to-end trainable model of photo acquisition and dissemination (Korus & Memon, 2019). The model uses a *forensic analysis network* (FAN) for photo manipulation detection, and allows for joint optimization of the FAN and the camera ISP, leading to distinct imaging artifacts that facilitate authentication. The published toolbox included only stan-

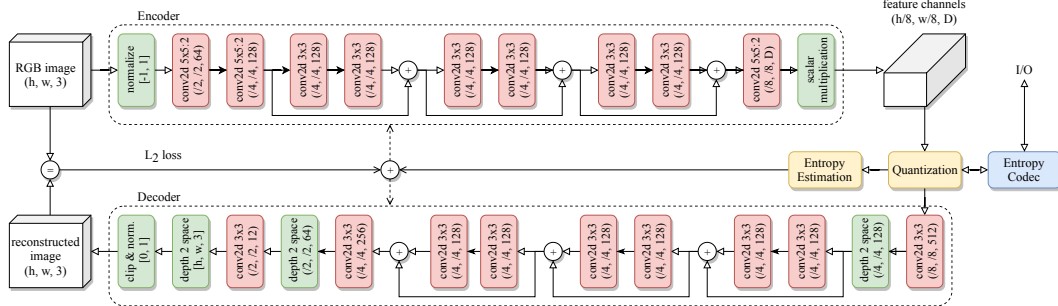

Figure 2: Architecture of our deep compression network: an auto-encoder with 3 sub-sampling stages and residual units in between. (Empty arrows: no activation; filled arrows: leaky ReLU.)

dard JPEG compression, and we extended it to support trainable codecs. We show a generic version of the updated model in Fig. 1 with highlighted potentially trainable elements. In this study, we fixed the camera model, and jointly optimize the FAN and a *deep compression network* (DCN). We describe the design of our DCN codec, and its pre-training protocol below.

## 3.1 BASELINE DCN ARCHITECTURE

Our DCN model follows the general auto-encoder architecture proposed by Theis et al. (2017), but uses different quantization, entropy estimation and entropy coding schemes (Section 3.2). The model is fully convolutional, and consists of 3 sub-sampling (stride-2) convolutional layers, and 3 residual blocks in between (Fig. 2). We do not use any normalization layers (such as GDN), and rely solely on a single trainable scaling factor. Distribution shaping occurs organically thanks to entropy regularization (see Fig. A.3b in the appendix). The decoder mirrors the encoder, and implements up-sampling using sub-pixel convolutions (combination of convolutional and depth-to-space layers).

We experimented with different variants of latent representation quantization, eventually converging on soft-quantization with a fixed codebook of integers with a given maximal number of bits per feature (bpf). We used a 5-bpf uniform codebook ($M = 32$ values from -15 to 16). We show the impact of codebook size in the appendix (Fig. A.3a).

The model is trained to minimize distortion between the input and reconstructed images regularized by entropy of the latent representation:

$$\mathcal{L}_{\text{dcn}} = \mathbb{E}_{\mathbf{X}} \left[ d\left(\mathbf{X}, \mathcal{D} \circ \mathcal{Q} \circ \mathcal{E}(\mathbf{X})\right) + \lambda_H H\left(\mathcal{Q} \circ \mathcal{E}(\mathbf{X})\right) \right], \quad (1)$$

where $\mathbf{X}$ is the input image, and $\mathcal{E}$, $\mathcal{Q}$, and $\mathcal{D}$ denote the encoder, quantization, and decoder, respectively. We used a simple $L_2$ loss in the RGB domain as the distortion measure $d(\cdot, \cdot)$, a differentiable soft estimate of entropy $H$ (Section 3.2), and SSIM as the validation metric.

## 3.2 SOFT QUANTIZATION AND ENTROPY ESTIMATION

We developed our own quantization and entropy estimation mechanism, because we found existing approaches unnecessarily complicated and/or lacking in accuracy. Some of the most recent solutions include: (1) addition of uniform random noise to quantized samples and non-parametric entropy modeling by a fitted piece-wise linear model (Ballé et al., 2016); (2) differentiable entropy upper bound with a uniform random noise component (Theis et al., 2017); (3) regularization by penalizing norm of quantized coefficients and differences between spatial neighbors (Rippel & Bourdev, 2017); (4) PixelCNN for entropy estimation and context modeling (Mentzer et al., 2018). Our approach builds upon the soft quantization used by Mentzer et al. (2018), but is extended to address numerical stability problems, and allow for accurate entropy estimation.

Let $\mathbf{z}$ be a vectorized latent representation $\mathbf{Z}$ of $N$ images, i.e.: $z_k = z_{n,i,j,c}$ where $n, i, j, c$ advance sequentially along an arbitrary memory layout (here image, width, height, channel). Let $\mathbf{c}$ denote a quantization codebook with $M$ centers $[c_1, \ldots, c_M]$ (code words). Then, given a weight matrix $\mathbf{W} \in [0,1]_{N,M} : \forall_m \sum_n w_{n,m} = 1$, we can define: *hard quantization* as $\hat{\mathbf{z}} = \left[ c_{\operatorname{argmax}_m \mathbf{w}_{:,m}} \right]$; and *soft*

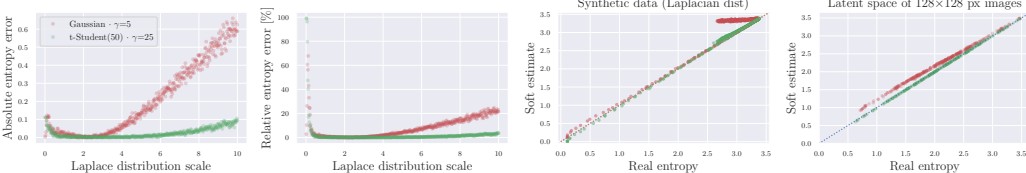

Figure 3: Entropy estimation error for a Laplacian distribution with varying scale and for the latent space of $128 \times 128$ px images. The t-Student kernel is significantly more accurate - especially for wide distributions overflowing the codebook range.

*quantization* as $\tilde{\mathbf{z}} = \mathbf{Wc}$. Hard quantization replaces an input value with the closest available code-word, and corresponds to a rounding operation performed by the image codec. Soft quantization is a differentiable relaxation, which uses a linear combination of all code-words - as specified by the weight matrix. A detailed comparison of both quantization modes, along with an illustration of potential numerical pitfalls, can be observed in the top row of Fig. A.1 in the appendix. The hard and soft quantization are used in the forward and backward passes, respectively. In Tensorflow, this can be implemented as $\mathbf{z}$ = `tf.stop_gradient(`$\hat{\mathbf{z}}$ - $\tilde{\mathbf{z}}$`)` + $\tilde{\mathbf{z}}$.

The weights for individual code-words in the mixture are computed by applying a kernel $\kappa$ to the distances between the values and the code-words, which can be organized into a distance matrix $\mathbf{D}$:

$$\mathbf{D} = \mathbf{z} - \mathbf{c}^{\mathsf{T}} = \left[ d_{n,m} = z_n - c_m \right] , \tag{2}$$

$$\mathbf{W} = \kappa(\mathbf{D}) = \left[ w_{n,m} = \kappa(d_{n,m}) \right] . \tag{3}$$

The most commonly used implementations use a Gaussian kernel:

$$\kappa_\gamma = e^{-\gamma d_{n,m}^2} , \tag{4}$$

which suffers from numerical problems for edge cases overflowing the codebook range (see Fig. A.1 top row in the 4-th and 5-th columns). To alleviate these problems, we adopt a t-Student kernel:

$$\kappa_{\gamma,v} = \left( 1 + \frac{\gamma d_{n,m}^2}{v} \right)^{-(v+1)/2} , \tag{5}$$

which behaves much better in practice. We do not normalize the kernels, and ensure correct proportions of the weights by numerically normalizing rows of the weight matrix.

We estimate entropy of the quantized values by summing the weight matrix along the sample dimension, which yields an estimate of the histogram w.r.t. codebook entries (comparison with an actual histogram is shown in Fig. A.3):

$$\tilde{\mathbf{h}} = \left[ \tilde{h}_m = \sum_n w_{n,m} \right] . \tag{6}$$

This allows to estimate the entropy of the latent representation by simply:

$$\hat{H} = - \sum_m \tilde{h}_m \log_2 \tilde{h}_m . \tag{7}$$

We assess the quality of the estimate both for synthetic random numbers (1,000 numbers sampled from Laplace distributions of various scales) and an actual latent representation of $128 \times 128$ px RGB image patches sampled from the *clic* test set (see Section 3.5 and examples in Fig. 4a). For the random sample, we fixed the quantization codebook to integers from $-5$ to $5$, and performed the experiment numerically. For the real patches, we fed the images through a pre-trained DCN model (a medium-quality model with 32 feature channels; 32-C) and used the codebook embedded in the model (integers from $-15$ to $16$).

Fig. 3 shows the entropy estimation error (both absolute and relative) and scatter plots of real entropies vs. their soft estimates using the Gaussian and t-Student kernels. It can be observed that the t-Student kernel consistently outperforms the commonly used Gaussian. The impact of the kernels' hyperparameters on the relative estimation error is shown in Fig. A.2. The best combination of kernel parameters ($v = 50, \gamma = 25$) is highlighted in red and used in all subsequent experiments.

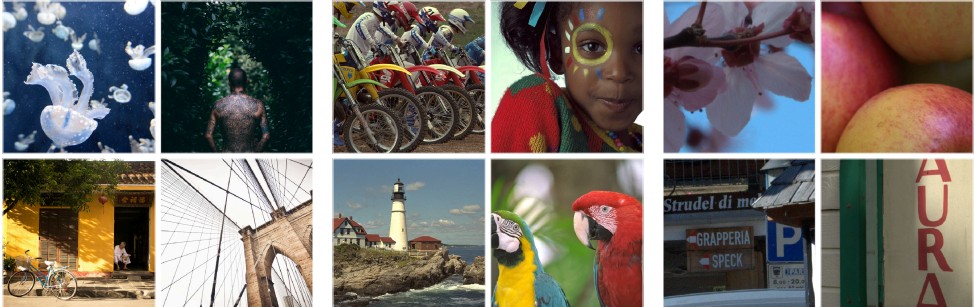

Figure 4: Example images from the considered *clic*, *kodak* and *raw* test sets (512×512 px).

### 3.3 ENTROPY CODING AND BIT-STREAM STRUCTURE

We used a state-of-the-art entropy coder based on asymmetric numeral systems (Duda, 2013; Duda et al., 2015) and its finite state entropy (FSE) implementation (Collet, 2013). For simplicity and computational efficiency, we did not employ a context model[1] and instead encode individual feature channels (*channel EC*). Bitrate savings w.r.t. global entropy coding (*global EC*) vary based on the model, image size and content. For $512 \times 512$ px images, we observed average savings of $\approx 12\%$, but for very small patches (e.g., 128 px), it may actually result in overhead (Tab. A.2). This can be easily addressed with a flag that switches between different compression modes, but we leave practical design of the format container for future work. We use a simple structure of the bit-stream, which enables variable-length, per-channel entropy coding with random channel access (Tab. A.1). Such an approach offers flexibility and scalability benefits, e.g.: (1) it allows for rapid analysis of selected feature channels (Torfason et al., 2018); (2) it enables trivial parallel processing of the channels to speed up encoding/decoding on modern multi-core platforms.

### 3.4 TRAINING PROTOCOL AND DATA

We pre-trained the DCN model in isolation and minimize the entropy-regularized $L_2$ loss (equation 1) on *mixed natural images* (MNI) from 6 sources: (1) native camera output from the RAISE and MIT-5k datasets (Dang-Nguyen et al., 2015; Bychkovsky et al., 2011); (2) photos from the Waterloo exploration database (Ma et al., 2016); (3) HDR images (Hasinoff et al., 2016); (4) computer game footage (Richter et al., 2016); (5) city scapes (Cordts et al., 2016); and (6) the training sub-set of the CLIC professional dataset (CLIC, 2019). In total, we collected 32,000 square crops ranging from $512 \times 512$ to $1024 \times 1024$ px, which were subsequently down-sampled to $256 \times 256$ px and randomly split into training and validation subsets.

We used three augmentation strategies: (1) we trained on $128 \times 128$ px patches randomly sampled in each step; (2) we flip the patches vertically and/or horizontally with probability 0.5; and (3) we apply random gamma correction with probability 0.5. This allowed for reduction of the training set size, to $\approx$10k images where the performance saturates. We used batches of 50 images, and learning rate starting at $10^{-4}$ and decaying by a factor of 0.5 every 1,000 epochs. The optimization algorithm was Adam with default settings (as of Tensorflow 1.12). We train by minimizing MSE ($L_2$ loss) until convergence of SSIM on a validation set with 1,000 images.

### 3.5 BASELINE MODELS AND EVALUATION

We control image quality by changing the number of feature channels. We consider three configurations for low, medium, and high quality with 16, 32, and 64 channels, respectively.

**Standard Codecs:** As hand-crafted baselines, we consider three codecs: JPEG from the libJPEG library via the *imageio* interface, JPEG2000 from the OpenJPEG library via the *Glymur* interface, and BPG from its reference implementation (Bellard, 2014). Since our model uses full-resolution

---

[1]Previous studies report $\approx$10% performance gains due to context modeling (Agustsson et al., 2018).

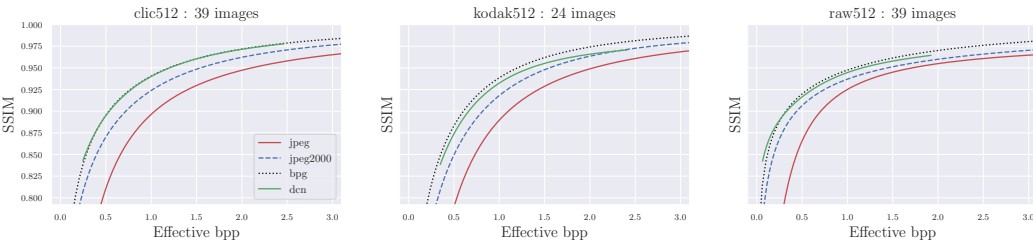

Figure 5: Rate-distortion trade-offs on the *clic*, *kodak* and *raw* test sets. (See Fig. A.5 for MS-SSIM).

RGB channels as input, we also use full-resolution chrominance channels whenever possible (JPEG and BPG). To make the comparison as fair as possible, we measure *effective payload* of the codecs. For the JPEG codec, we manually seek byte markers and include only the Huffman tables and Huffman-coded image data. For JPEG2000, we add up lengths of tile-parts, as reported by *jpylyzer*. For BPG, we seek the `picture_data_length` marker.

**Rate-distortion Trade-off:** We used 3 datasets for the final evaluation (Fig. 4): (*raw*) 39 images with native camera output from 4 different cameras (Dang-Nguyen et al., 2015; Bychkovsky et al., 2011); (*clic*) 39 images from the professional validation subset of CLIC (2019); (*kodak*) 24 images from the standard Kodak dataset. All test images are of size $512 \times 512$px, and were obtained either by cropping directly without re-sampling (*raw, kodak*) or by resizing a central square crop (*clic*).

We measured PSNR and SSIM using the *scikit-image* package and MS-SSIM using *sewar*. We used the default settings and only set the data range to $[0, 1]$. The values are computed as the average over RGB channels. The bit-rate for hand-crafted codecs was computed using the *effective payload*, as explained above. For the DCN codec, we completely encoded and decoded the images (Section A.1).

Fig. 5 shows rate-distortion curves (SSIM vs. effective bpp) for the *clic* and *raw* datasets (see appendix for additional results). We show 4 individual images (Fig. 4) and averages over the respective datasets. Since standard quality control (e.g., quality level in JPEG, or number of channels in DCN) leads to a wide variety of bpps, we fit individual images to a parametric curve $f(x) = (1 + e^{-\alpha x^\beta + \gamma})^{-1} - \delta$ and show the averaged fits. It can be observed that our DCN model delivers significantly better results than JPEG and JPEG2000, and approaches BPG.

**Processing Time:** We collected DCN processing times for various platforms (Table 1), including desktops, servers, and low-power edge AI. We report network inference and complete encoding/decoding times on $512 \times 512$ px and $1920 \times 1080$ px images, averaged over the *clic* dataset (separate runs with batch size 1) and obtained using an unoptimized Python 3 implementation[2]. On GPU-enabled platforms, the inference time becomes negligible (over 100 fps for $512 \times 512$ px images, and over 20 fps for $1920 \times 1080$ px images on GeForce 1080 with a 2.6 GHz Xeon CPU), and entropy coding starts to become the bottleneck (down to 13 and 5 fps, respectively). We emphasize that the adopted FSE codec is one of the fastest available, and significantly outperforms commonly used arithmetic coding (Duda, 2013). If needed, *channel EC* can be easily parallelized, and the ANS codec could be re-implemented to run on GPU as well (Weißenberger & Schmidt, 2019).

As a reference, we measured the processing times of $1920 \times 1080$ px images for the standard codecs on the i7-7700 CPU @ 3.60GHz processor. JPEG coding with 1 thread takes between 0.061 s (Q=30) and 0.075 s (Q=90) (inclusive of writing time to RAM disk; using the *pillow* library). JPEG 2000 with 1 thread takes 0.61 s regardless of the quality level (inclusive of writing time to RAM disk; *glymur* library). BPG with 4 parallel threads takes 2.4 s (Q=1), 1.25 s (Q=20) and 0.72 s (Q=30) (inclusive of PNG reading time from RAM disk; *bpgenc* command line tool). While not directly comparable and also not optimized, some state-of-the-art deep learned codecs require minutes to process even small images, e.g., 5-6 min for $768 \times 512$ px images from the Kodak dataset reported by Mentzer et al. (2018). The fastest state-of-the-art learned codec is reported to run at $\approx$100 fps on images of that size on a GPU-enabled desktop computer (Rippel & Bourdev, 2017).

---

[2]We used low-level Cython wrappers for the FSE entropy coder (Collet, 2013).

Table 1: Average compression/decompression time on different platforms (in seconds) with breakdown into NN inference and complete processing; in-memory processing using the 32-C model.

| GPU | CPU / Platform | 512×512 px images | | | | 1920×1080 px images | | | |
| | | inference | | whole codec | | inference | | whole codec | |
| | | Encode | Decode | Encode | Decode | Encode | Decode | Encode | Decode |
| --- | --- | --- | --- | --- | --- | --- | --- | --- | --- |
| Maxwell | ARM 57 (nVidia Jetson Nano) | 0.2076 | 0.5333 | 0.6507 | 0.6721 | 1.6348 | 3.6057 | 4.4978 | 4.9722 |
| - | i7-7700 @ 3.60GHz | 0.2165 | 0.3330 | 0.2272 | 0.3317 | 1.8052 | 2.7678 | 1.8753 | 2.7901 |
| - | i7-9770 @ 3.60Ghz | 0.0648 | 0.1396 | 0.0762 | 0.1397 | 0.5197 | 1.1685 | 0.6080 | 1.1728 |
| GF 1080 | Xeon E5-2690 @ 2.60GHz | 0.0083 | 0.0173 | 0.0742 | 0.0498 | 0.0597 | 0.1244 | 0.1805 | 0.1714 |
| P40 | Xeon E5-2680 @ 2.40GHz | 0.0093 | 0.0160 | 0.0720 | 0.0375 | 0.0558 | 0.1123 | 0.1895 | 0.1684 |
| V100 | Xeon E5-2680 @ 2.40GHz | 0.0065 | 0.0071 | 0.0604 | 0.0209 | 0.0416 | 0.0489 | 0.1735 | 0.0979 |
| GF 2080S | i7-9770 @ 3.60Ghz | 0.0059 | 0.0132 | 0.0421 | 0.0244 | 0.0399 | 0.0953 | 0.1343 | 0.1320 |

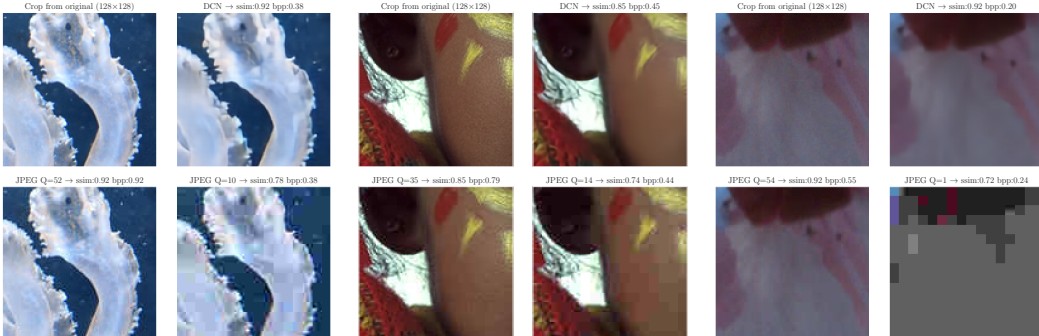

Figure 6: Comparison of our DCN codec with low-quality settings (16-C) against JPEG with matching SSIM and matching bpp. Samples from *clic*, *kodak*, and *raw* datasets.

## 4 OPTIMIZATION FOR MANIPULATION DETECTION

We consider the standard photo manipulation detection setup where an adversary uses content-preserving post-processing, and a forensic analysis network (FAN) needs to identify the applied operation or confirm that the image is unprocessed. We use a challenging real-world setup, where the FAN can analyze images only after transmission through a lossy dissemination channel (Fig. 1). In such conditions, forensic analysis is known to fail (Korus & Memon, 2019). We consider several versions of the channel, including: standard JPEG compression, pre-trained DCN codecs, and trainable DCN codecs jointly optimized along with the FAN. We analyze $128 \times 128$ px patches, and don't use down-sampling to isolate the impact of the codec.

### 4.1 PHOTO MANIPULATION AND DETECTION STRATEGY

We consider 6 benign post-processing operations which preserve image content, but change low-level traces that can reveal a forgery. Such operations are commonly used either during photo manipulation or as a masking step afterwards. We include: (a) *sharpening* - implemented as an unsharp mask operator applied to the luminance channel in the HSV color space; (b) *resampling* involving successive down-sampling and up-sampling using bilinear interpolation and scaling factors 1:2 and 2:1; (c) *Gaussian filtering* with a $5 \times 5$ filter and standard deviation 0.83; (d) *JPEG compression* using a differentiable *dJPEG* model with quality level 80; (e) *AWGN* noise with standard deviation 0.02; and (f) *median* filtering with a $3 \times 3$ kernel. The operations are difficult to distinguish visually from native camera output - even without lossy compression (Fig. 7).

The FAN is a state-of-the-art image forensics CNN with a constrained residual layer (Bayar & Stamm, 2018). We used the model provided in the toolbox (Korus & Memon, 2019), and optimize for classification (native camera output + 6 post-processing classes) of RGB image patches. In total, the model has 1.3 million parameters.

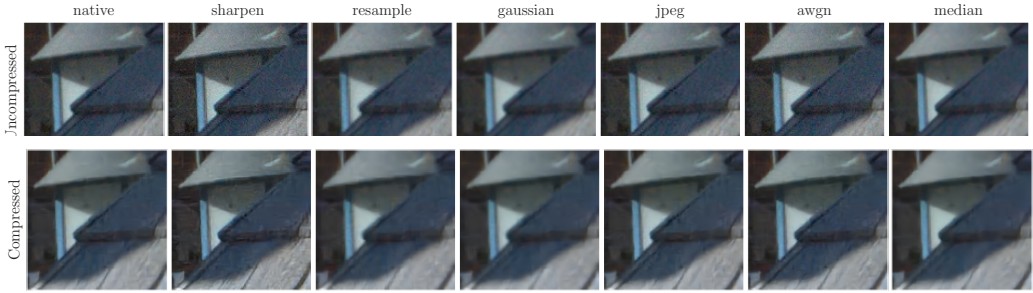

Figure 7: Subtle photo manipulation: (left) 128×128 px native patch; (rest) various post-processing.

## 4.2 TRAINING PROTOCOL

We jointly train the entire workflow and optimize both the FAN and DCN models. Let $\mathcal{M}_c$ denote the $c$-th manipulation (including identity for native camera output), and $\mathcal{F}$ denote the output of the FAN with $\mathcal{F}_c$ being the probability of the corresponding manipulation class $c$. Let also $\mathcal{C}$ denote the adopted lossy compression model, e.g., $\mathcal{D} \circ \mathcal{Q} \circ \mathcal{E}$ for the DCN. We denote sRGB images rendered by the camera ISP as $\mathbf{X}$. The FAN model is trained to minimize a cross-entropy loss:

$$\mathcal{L}_{\text{ce}} = \mathop{\mathbb{E}}_{\mathbf{X}} \left[ \sum_{c=1}^{7} \log \left( \mathcal{F}_c \circ \mathcal{C} \circ \mathcal{M}_c(\mathbf{X}) \right) \right] , \tag{8}$$

and the DCN to minimize its combination with the original fidelity/entropy loss (equation 1):

$$\mathcal{L} = \mathcal{L}_{\text{ce}} + \lambda_c \mathcal{L}_{\text{dcn}} , \tag{9}$$

where $\lambda_c$ is used to control the balance between the objectives (we consider values from $10^{-3}$ to 1). We start from pre-trained DCN models (Section 3.4). The FAN model is trained from scratch.

When JPEG compression was used in the channel, we used the differentiable *dJPEG* model from the original study (Korus & Memon, 2019), but modified it to use hard-quantization in the forward pass to ensure results equivalent to libJPEG. We used quality levels from 10 to 95 with step 5.

We followed the same training protocol as Korus & Memon (2019), and trained on *native camera output* (NCO). We used the *DNet* pipeline for Nikon D90, and randomly sampled $128 \times 128$ px RGB patches from 120 full-resolution images. The remaining 30 images were used for validation (we sampled 4 patches per image to increase diversity). We used batches of 20 images, and trained for 2,500 epochs with learning rate starting at $10^{-4}$ and decaying by 10% every 100 epochs. For each training configuration, we repeated the experiment 3-5 times to validate training stability.

## 4.3 QUANTITATIVE ANALYSIS

We summarize the obtained results in Fig. 8 which shows the trade-off between effective bpp (rate), SSIM (distortion), and manipulation detection accuracy. The figure compares standard JPEG compression (diamond markers), pre-trained basic DCN models (connected circles with black border), and fine-tuned DCN models for various regularization strenghts $\lambda_c$ (loose circles with gray border). Fine-tuned models are labeled with a delta in the auxiliary metric (also encoded as marker size and color), and the text is colored in red/green to indicate deterioration or improvement.

Fig. 8a shows how the manipulation detection capability changes with effective bitrate of the codec. We can make the following observations. Firstly, JPEG delivers the worst trade-off and exhibits irregular behavior. The performance gap may be attributed to: (a) better image fidelity for the DCN codec, which retains more information at any bitrate budget; (b) presence of JPEG compression as one of the manipulations. The latter factor also explains irregular drops in accuracy, which coincide with the quality level of the manipulation (80) and unfavorable multiples of the quantization tables (see also Fig. B.1). Secondly, we observe that fine-tuning the DCN model leads to a sudden increase in payload requirements, minor improvement in quality, and gradual increase in manipulation detection accuracy (as $\lambda_c \to 0$). We obtain significant improvements in accuracy even for the lowest

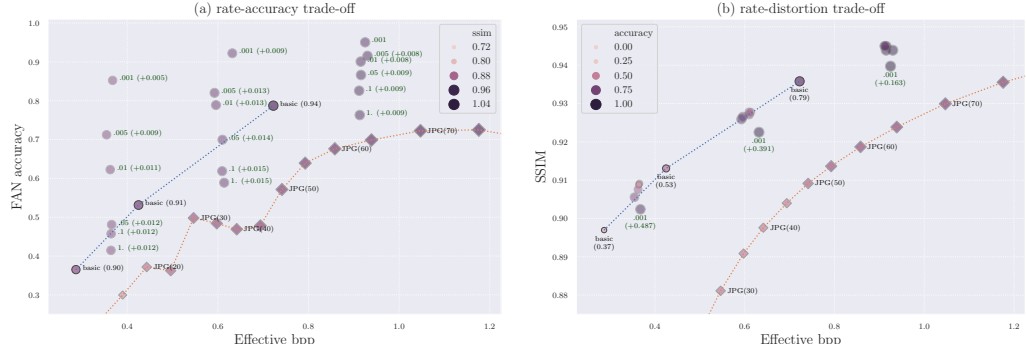

Figure 8: The rate-distortion-accuracy trade-off (*raw* test set) reveals significant improvements in manipulation detection accuracy while maintaining similar rate-distortion performance.

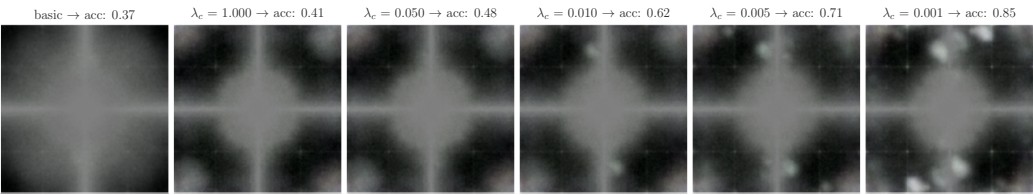

Figure 9: Visualization of frequency attenuation/amplification patterns in the FFT domain for the fine-tuned DCN codec (low-quality, 16-C model).

quality level ($37\% \rightarrow 85\%$; at such low bitrates JPEG stays below $30\%$). Interestingly, we don't see major differences in payload between the fine-tuned models, which suggests that qualitative differences in encoding may be expected beyond simple inclusion of more information.

Fig. 8b shows the same results from a different perspective, and depicts the standard rate-distortion trade-off supplemented with auxiliary accuracy information. We can observe that DCN fine-tuning moves the model to a different point (greater payload, better quality), but doesn't seem to visibly deteriorate the rate-distortion trade-off (with the exception of the smallest regularization $\lambda_c = 0.001$ which consistently shows better accuracy and worse fidelity).

## 4.4 QUALITATIVE ANALYSIS

To explain the behavior of the models, we examine frequency attenuation patterns. We compute FFT spectra of the compressed images, and divide them by the corresponding spectra of uncompressed images. We repeat this procedure for all images in our *raw* test set, and average them to show consistent trends. The results are shown in Fig. 9 for the pre-trained DCN model (1st column) and fine-tuned models with decreasing $\lambda_c$ (increasing emphasis on accuracy). The plots are calibrated to show unaffected frequencies as gray, and attenuated/emphasized frequencies as dark/bright areas.

The pre-trained models reveal clear and gradual attenuation of high frequencies. Once the models are plugged in to the dissemination workflow, high frequencies start to be retained, but it does not suffice to improve manipulation detection. Increasing importance of the cross-entropy loss gradually changes the attenuation patterns. Selection of frequencies becomes more irregular, and some bands are actually emphasized by the codec. The right-most column shows the most extreme configuration where the trend is clearly visible (the outlier identified in quantitative analysis in Section 4.3).

The changes in codec behavior generally do not introduce visible differences in compressed images (as long as the data distribution is similar, see Section 5). We show an example image (*raw* test set), its compressed variants (low-quality, 16-C), and their corresponding spectra in Fig. 10. The spectra follow the identified attenuation pattern (Fig. 9). The compressed images do not reveal any obious artifacts, and the most visible change seems to be the jump in entropy, as predicted in Section 4.3.

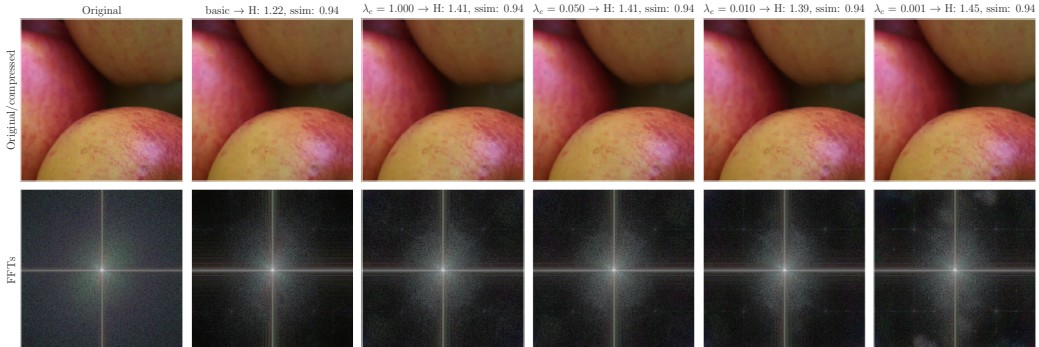

Figure 10: Compression results for various versions of the low-quality DCN: (1st column) original image; (2nd) pre-trained model; (3rd-6th) fine-tuned models with decreasing $\lambda_c$.

## 5   DISCUSSION, LIMITATIONS AND FUTURE WORK

While the proposed approach can successfully facilitate pre-screening of photographs shared online, further research is needed to improve model generalization. We observed that the fine-tuning procedure tends bias the DCN/FAN models towards the secondary image dataset, in our case the native camera output (NCO). The baseline DCN was pre-trained on mixed natural images (MNI) with extensive augmentation, leading to competitive results on all test sets. However, fine-tuning was performed on NCO only. Characteristic pixel correlations, e.g., due to color interpolation, bias the codec and lead to occasional artifacts in MNIs (mostly in the *clic* test set; see Appendix B), and deterioration of the rate-distortion trade-off. The problem is present regardless of $\lambda_c$, which suggests issues with the fine-tuning protocol (data diversity) and not the forensic optimization objective.

We ran additional experiments by skipping photo acquisition and fine-tuning directly on MNI from the original training set (subset of 2,500 RGB images). We observed the same behavior (see Appendix C), and the optimized codec was artifact-free on all test sets. (Although, due to a smaller training set, the model loses some of its performance; cf. MNI results in Fig. A.6.) However, the FANs generalized well only to *clic* and *kodak* images. The originally trained FANs generalized reasonably well to different NCO images (including images from other 3 cameras) but not to *clic* or *kodak*. This confirms that existing forensics models are sensitive to data distribution, and that further work will be needed to establish more universal training protocols (see detailed discussion in Appendix D). Short fine-tuning is known to help (Cozzolino et al., 2018), and we leave this aspect for future work. We are also planning to explore new transfer learning protocols (Li & Hoiem, 2017).

Generalization should also consider other forensic tasks. We optimized for manipulation detection, which serves as a building block for more complex problems, like processing history analysis or tampering localization (Korus, 2017; Mayer & Stamm, 2019; Wu et al., 2019; Marra et al., 2019a). However, additional pre-screening may also be needed, e.g., analysis of sensor fingerprints (Chen et al., 2008), or identification of computer graphics or synthetic content (Marra et al., 2019b).

## 6   CONCLUSIONS

Our study shows that lossy image codecs can be explicitly optimized to retain subtle low-level traces that are useful for photo manipulation detection. Interestingly, simple inclusion of high frequencies in the signal is insufficient, and the models learns more complex frequency attenuation/amplification patterns. This allows for reliable authentication even at very low bit-rates, where standard JPEG compression is no longer practical, e.g., at bit-rates around 0.4 bpp where our DCN codec with low-quality settings improved manipulation detection accuracy from 37% to 86%. We believe the proposed approach is particularly valuable for online media platforms (e.g., Truepic, or Facebook), who need to pre-screen content upon reception, but need to aggressively optimize bandwidth/storage.

**Source Code:** `github.com/pkorus/neural-imaging`.

**Acknowledgements:** This work was supported by the NSF award number 1909488.

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

Table A.1: Structure of the bit-stream describing a DCN-compressed image

| Section | Content | Data Type | Bytes |
|---|---|---|---|
| Basic meta-data: | Latent shape H x W x N | uint8 | 3 |
| | Length of coded channel sizes = 2 bytes (uint16) | uint16 | 2 |
| Channel sizes (shorter of a/b) | (a) FSE-encoded channel sizes[1] | uint16 | var |
| | (b) raw bytes | uint16 | 2N |
| Image data ($N\times$ shorter of a/b) | (a) FSE-encoded latent channel[1] | uint8 | var |
| | (b) RLE-encoded latent channel (#repetitions + byte) | uint16 + uint8 | 3 |

[1] - inclusive of both ANS probability tables and entropy-coded data

Table A.2: Bit-stream length of *channel entropy coding (EC)* relative to *global EC* for different quality levels and image patches of various size.

| DCN model | Avg. bit-stream size | | | Bit-stream size range | | |
|---|---|---|---|---|---|---|
| | 128 | 256 | 512 | 128 | 256 | 512 |
| low quality (16-C) | 1.03 | 0.934 | 0.882 | 0.917 - 1.098 | 0.829 - 1.005 | 0.755 - 0.980 |
| medium quality (32-C) | 1.05 | 0.933 | 0.874 | 0.961 - 1.108 | 0.821 - 0.998 | 0.742 - 0.968 |
| high quality (64-C) | 1.07 | 0.948 | 0.887 | 0.977 - 1.119 | 0.833 - 0.998 | 0.773 - 0.964 |

# A  DCN CODEC DETAILS

## A.1  QUANTIZATION AND ENTROPY REGULARIZATION

The standard soft quantization with a Gaussian kernel (Mentzer et al., 2018) works well for rounding to arbitrary integers, but leads to numerical issues for smaller codebooks. Values significantly exceeding codebook endpoints have zero affinity to any of the entries, and collapse to the mean (i.e., $\approx 0$ in our case; Fig. A.1a). Such issues can be addressed by increasing numerical precision, sacrificing accuracy (due to larger kernel bandwidth), or adding explicit conditional statements in the code. The latter approach is inelegant and cumbersome in graph-based machine learning frameworks like Tensorflow. We used a t-Student kernel instead and increased precision of the computation to 64-bits. This doesn't solve the problem entirely, but successfully eliminated all issues that we came across in our experiments, and further improved our entropy estimation accuracy. Fig. A.2 shows entropy estimation error for Laplace-distributed random values, and different hyper-parameters of the kernels. We observed the best results for a t-Student kernel with 50 degrees of freedom and bandwidth $\gamma = 25$ (marked in red). This kernel is used in all subsequent experiments.

We experimented with different codebooks and entropy regularization strengths. Fig. A.3a shows how the quantized latent representation (QLR) changes with the size of the codebook. The figures also compare the actual histogram with its soft estimate (equation 6). We observed that the binary codebook is sub-optimal and significantly limits the achievable image quality, especially as the number of feature channels grows. Adding more entries steadily improved quality and the codebook with $M = 32$ entires (values from -15 to 16) seemed to be the point of diminishing returns.

Our entropy-based regularization turned out to be very effective at shaping the QLR (Fig. A.3b) and dispensed with the need to use other normalization techniques (e.g., GDN). We used only a single scalar multiplication factor responsible for scaling the distribution. All baseline and fine-tuned models use $\lambda_H = 250$ (last column). Fig. A.4 visually compares the QLRs of our baseline low-quality codec (16 feature channels) with weak and strong regularization.

## A.2  ENTROPY CODING AND BIT-STREAM STRUCTURE

We used a rudimentary bit-stream structure with the essential meta-data and markers that allow for successful decoding (Tab. A.1). Feature channels are entropy-coded independently (we refer to this approach as *channel EC*), and can be accessed randomly after decoding a lookup table of their sizes. This simple approach can yield considerable savings w.r.t. global entropy coding (*global EC*), especially as the image size increases (Tab. A.2).

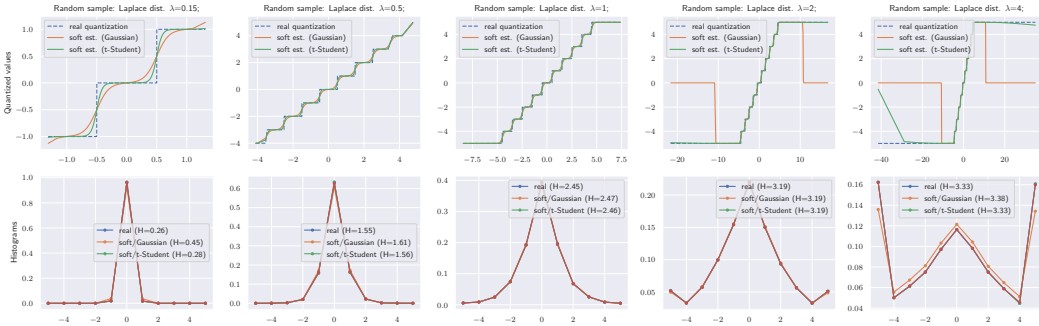

Figure A.1: Comparison of soft quantization with Gaussian and t-Student kernels for a Laplace distribution of increasing scale: the t-Student kernel is more accurate and robust to outliers.

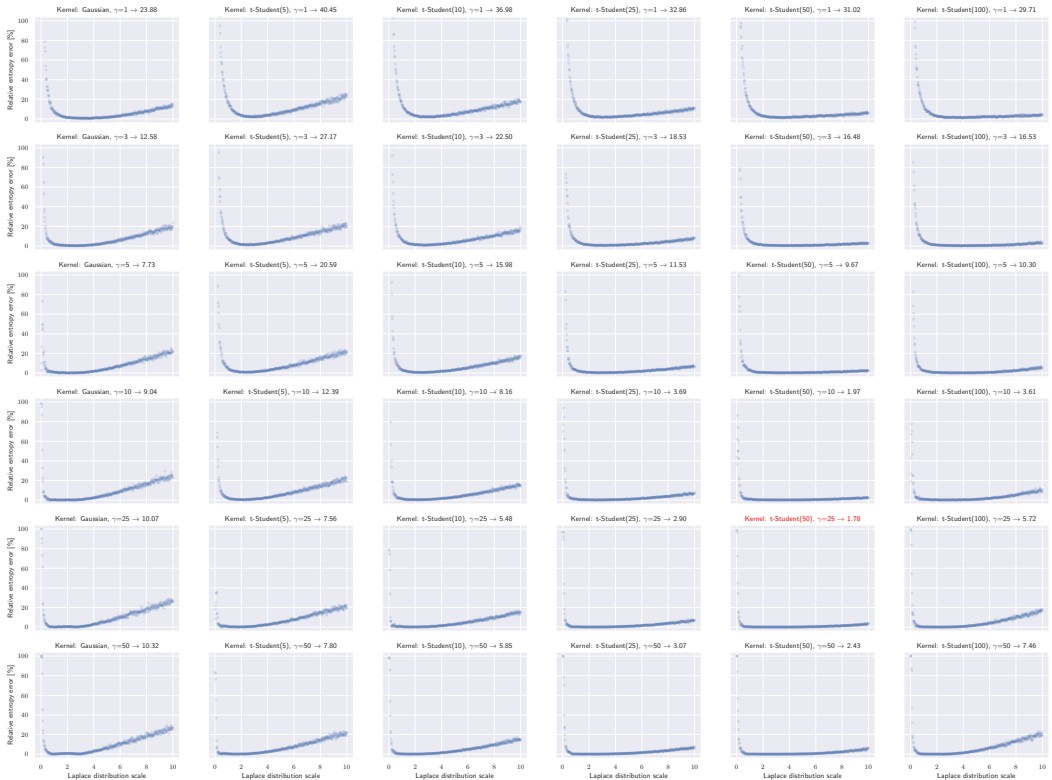

Figure A.2: Entropy estimation error for a Laplacian distribution with varying scale and various hyper-parameters of the kernels: the t-Student kernel (2nd-6th column) is more accurate than the Gaussian (1st column) - especially for wide distributions overflowing the codebook range.

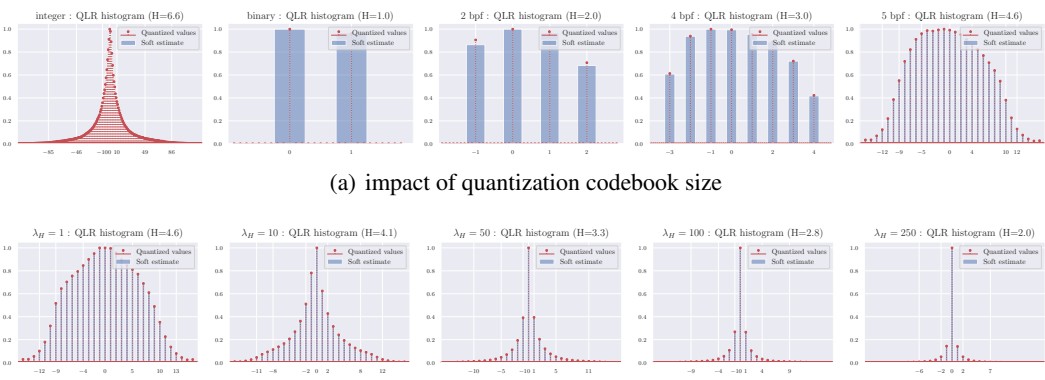

Figure A.3: Comparison of the distributions of the learned quantized latent representations (QLRs): (a) impact of codebook size; (b) impact of entropy regularization.

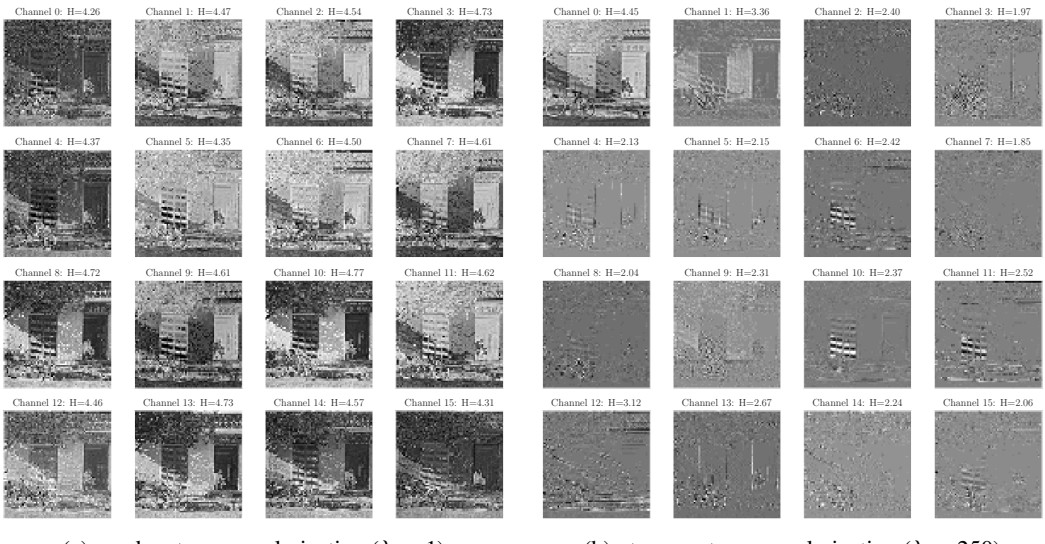

Figure A.4: Comparison of the latent representations (16-C model) of an example $512 \times 512$ px image (thong-vo-428) learned with weak and strong entropy regularization (5-bpf codebook).

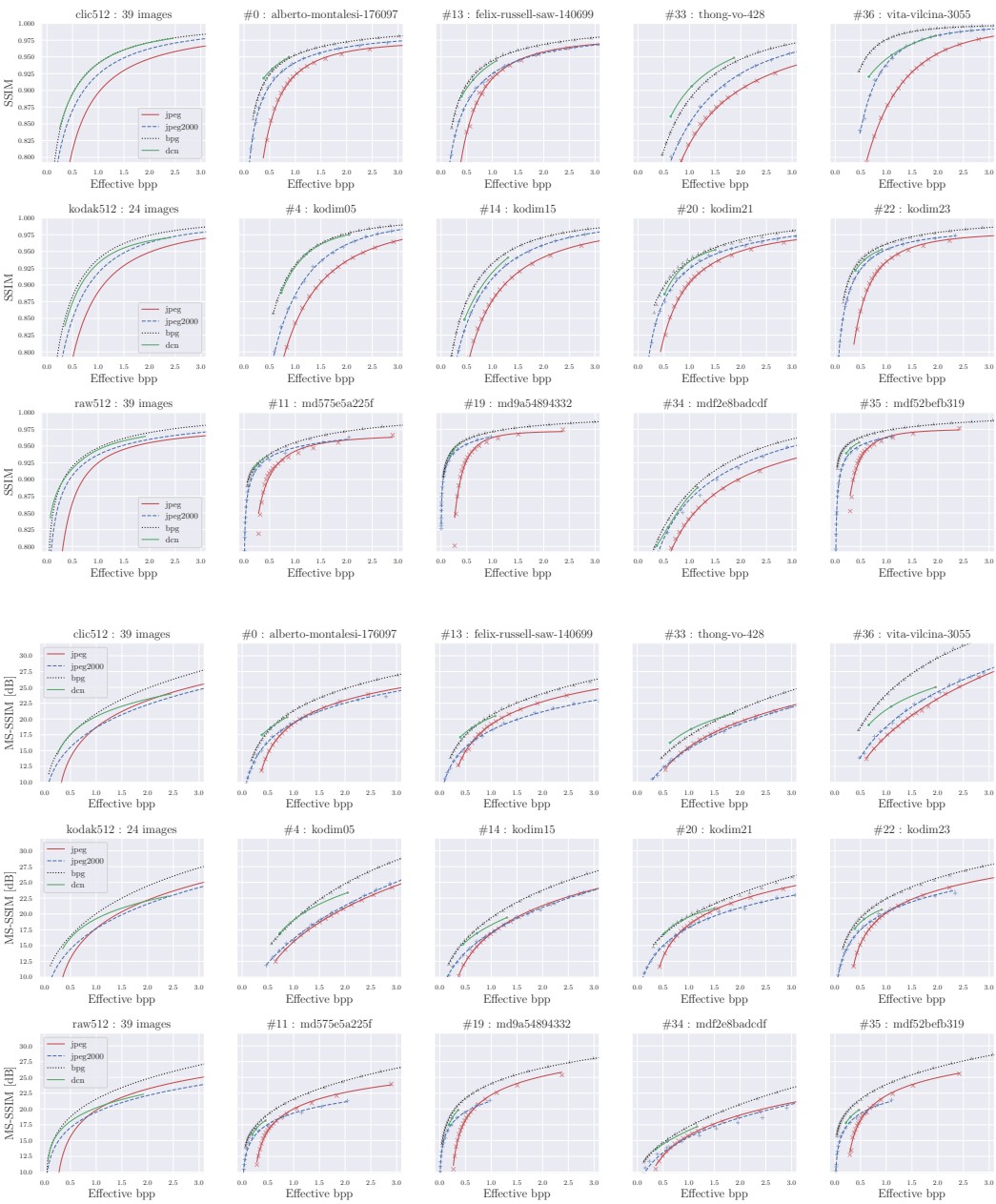

Figure A.5: Rate-distortion trade-offs of the baseline DCNs on the *clic*, *kodak* and *raw* test sets: (left) average over all images; (2nd-5th columns) sample images from Fig. 4; (top section) linear-scale SSIM; (bottom section) MS-SSIM in dB scale $f(x) = -10 \cdot \log_{10}(1 - x)$.

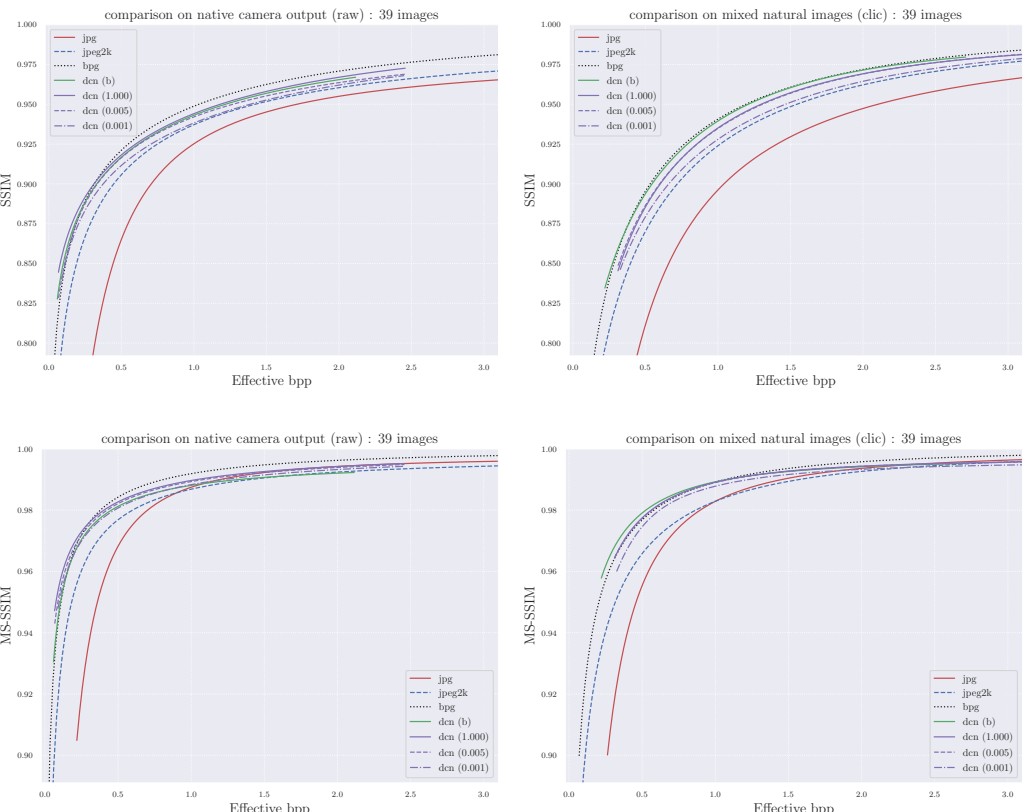

Figure A.6: Rate-distortion performance for standard codecs and all DCN versions including the pre-trained baselines (b) and 3 fine-tuned models with the the weakest ($\lambda_c = 1$) and the strongest emphasis on manipulation detection ($\lambda_c = 0.005$ and $0.001$). The latter incure only fractional cost in payload/quality but bring significant benefits for manipulation detection. The top (bottom) rows show results for SSIM (MS-SSIM), respectively, and include DCN models fine tuned on native camera output (*raw*) and mixed natural images (*clic*).

Table B.1: Example confusion matrices for the baseline and fine-tuned low-quality DCN models.

| True \ Predicted | (a) pre-trained 16-C DCN → 36.9% | | | | | | | (b) fine-tuned w. $\lambda_c = 0.0010$ → 87.0% | | | | | | |
| --- | --- | --- | --- | --- | --- | --- | --- | --- | --- | --- | --- | --- | --- | --- |
| | native | sharpen | resample | gaussian | jpeg | awgn | median | native | sharpen | resample | gaussian | jpeg | awgn | median |
| native | 8 | 10 | 10 | 24 | 7 | 18 | 23 | 92 | * | | * | * | * | * |
| sharpen | 8 | 46 | 8 | 3 | 5 | 18 | 12 | 7 | 86 | | | 8 | | |
| resample | * | | 58 | 25 | * | 5 | 8 | | | 88 | 11 | | * | |
| gaussian | * | | 12 | 65 | * | 7 | 15 | | | * | 95 | | | * |
| jpeg | 12 | 9 | 10 | 22 | 11 | 16 | 20 | * | * | | 3 | 91 | | * |
| awgn | 12 | 12 | 11 | 9 | 5 | 41 | 10 | * | 4 | * | | | 94 | |
| median | 4 | * | 8 | 45 | 3 | 9 | 29 | * | | 6 | 28 | | * | 63 |

# B    FINE-TUNING ON NATIVE CAMERA OUTPUT

**Controlling Detection Accuracy:**    Fig. 8 visualizes the trade-offs in image compression and foren-sic analysis performance. Here we show direct impact of image compression and fine-tuning settings on the achievable manipulation detection accuracy and its variations (Fig. B.1). For the JPEG codec, we observe nearly perfect manipulation detection for the highest quality level (QF=95), and a steady decline starting immediately below. The sudden drop in accuracy corresponds to the quality level of JPEG as one of the manipulations (QF=80). For DCN models, we clearly see steady improvement of fine-tuning w.r.t. the baseline models (on the right). Interestingly, the high-quality model shows a slight decline at first.

**Qualitative Analysis:**    The learned frequency attenuation/amplification patterns for all of the con-sidered quality levels are shown in Fig. B.2. The visualizations were computed in the FFT domain and show the relative magnitude of individual frequencies w.r.t. original uncompressed images (av-eraged over all test images). In all cases, we observe complex behavior beyond simple inclusion of high-frequency content. The pattern seem to have a stable trajectory, despite independent runs of the experiment with different regularization strengths $\lambda_c$. The same patterns will also be visible in individual image spectra (Fig. B.4 - Fig. B.6).

**Generalization and Imaging Artifacts:**    While our baseline DCN models were pre-trained on a large and diverse training set, the fine-tuning procedure relied on the complete model of photo acquisition and dissemination. Photo acquisition with digital cameras yields characteristic imaging artifacts, e.g., due to color filtering and interpolation. This leads to a characteristic distribution of native camera output (NCO), and ultimately biases the codec. Fig. B.3 shows the differences in SSIM between the baseline models and the models fine-tuned with a very weak cross-entropy objective (leading to no improvement in manipulation detection accuracy). For NCO (*raw* test set), we observe improvement of image quality (and corresponding increase in bitrate). For the *kodak* set, the quality remains mostly unaffected (with an increased bitrate). On the *clic* set, we observe minor quality loss, and occasional artifacts (see examples in Fig. B.4).

In Fig. B.4 - Fig. B.6 we collected example images from all test sets (*clic*, *kodak*, and *raw*) and compress them with baseline and fine-tuned models. The images are ordered by SSIM deterioration due to weak fine-tuning (quality loss without gains in accuracy; Fig. B.3) - the worst cases are shown at the top. (Note that some of the artifacts are caused by JPEG encoding of the images embedded in the PDF, and some geometric distortions were introduced by imperfect scaling in *matplotlib*.) In the first *clic* image (1st row in Fig. B.4), we can see color artifacts along the high-contrast wires. In the second image, we see distortions in the door blinds, and a subtle shift in the hue of the bike. For the remaining two images, SSIM remains the same or better and we do not see any artifacts. In the *kodak* set, the worst image quality was observed for *kodim05* (1st row in Fig. B.5), but we don't see any artifacts.

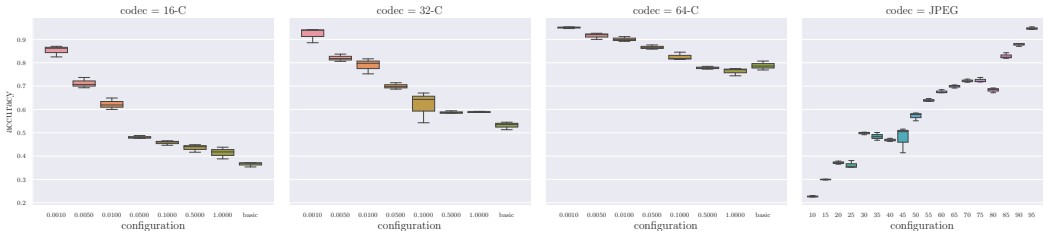

Figure B.1: Impact of compression quality and fine-tuning regularization on the achievable detection accuracy and its variation. Sudden drops for JPEG are caused by inclusion of this compression as one of the manipulations, and correspond to the manipulation quality level (80).

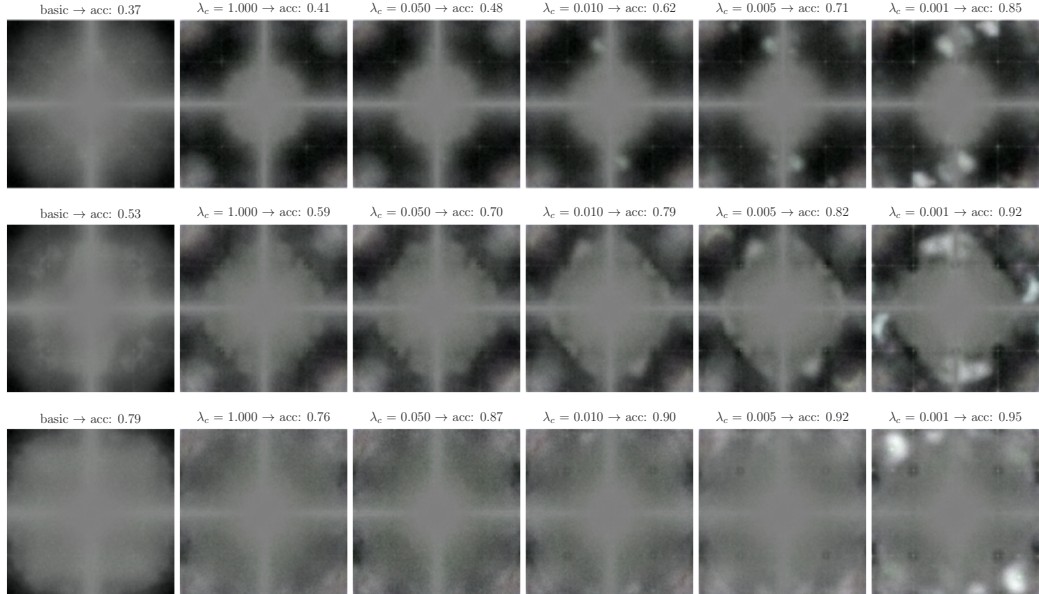

Figure B.2: Visualization of frequency attenuation/amplification patterns in the FFT domain for the fine-tuned DCN codecs (on native camera output). From the top: 16-C, 32-C, and 64-C models.



Figure B.3: Difference in image fidelity (SSIM) after fine-tuning the low-quality DCN model within the dissemination workflow (weak CE objective with little to no improvement in detection accuracy): *raw*, *kodak* and *clic* datasets.

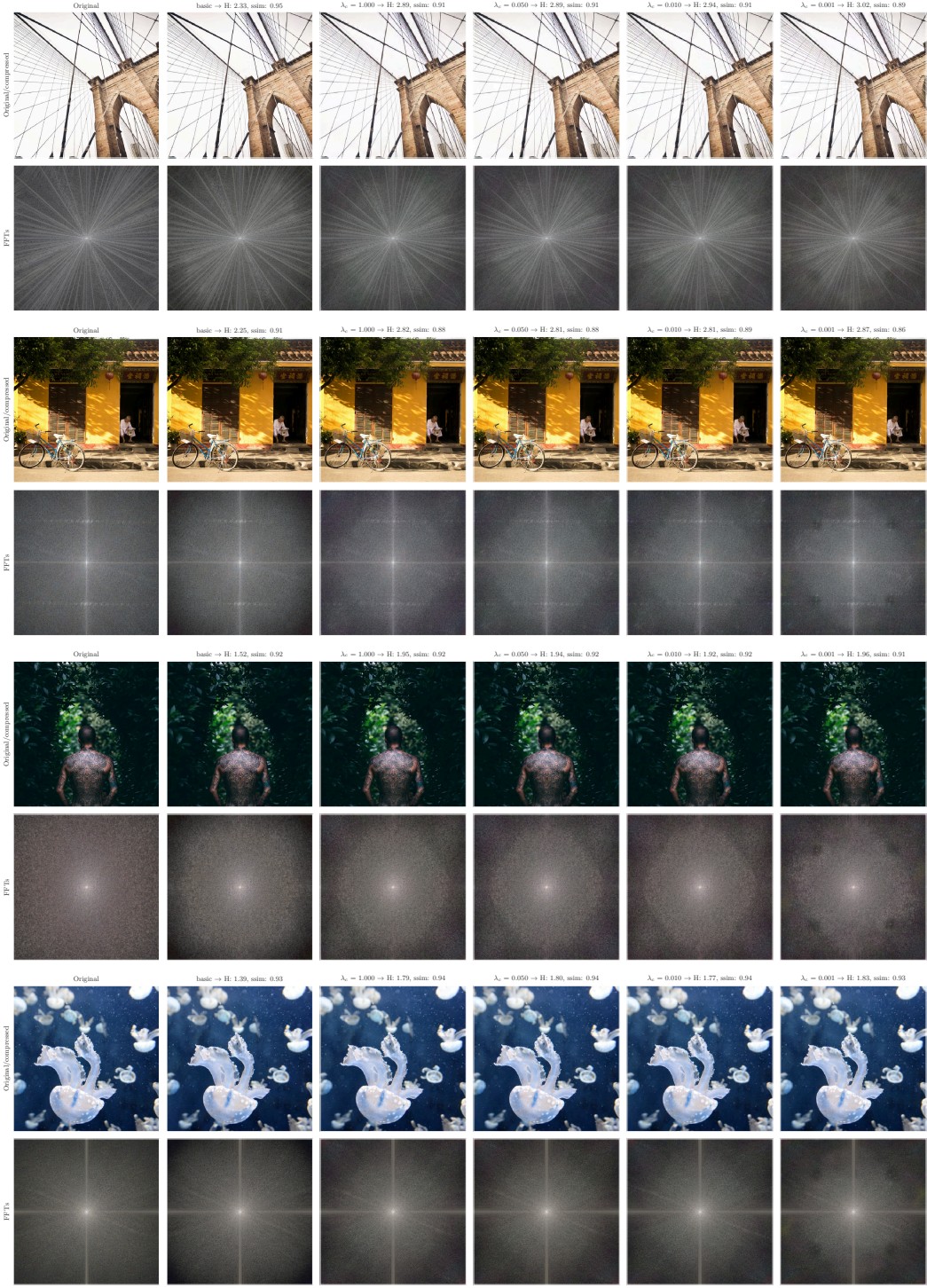

Figure B.4: Changes in the image compressed with various versions of the medium-quality DCN codec: (1st column) sample image from the *clic* dataset; (2nd) pre-trained DCN model; (3rd-6th) fine-tuned models with decreasing $\lambda_c$. Images are ordered by SSIM fidelity loss of $\lambda_c = 1.0$ w.r.t. the pre-trained model.

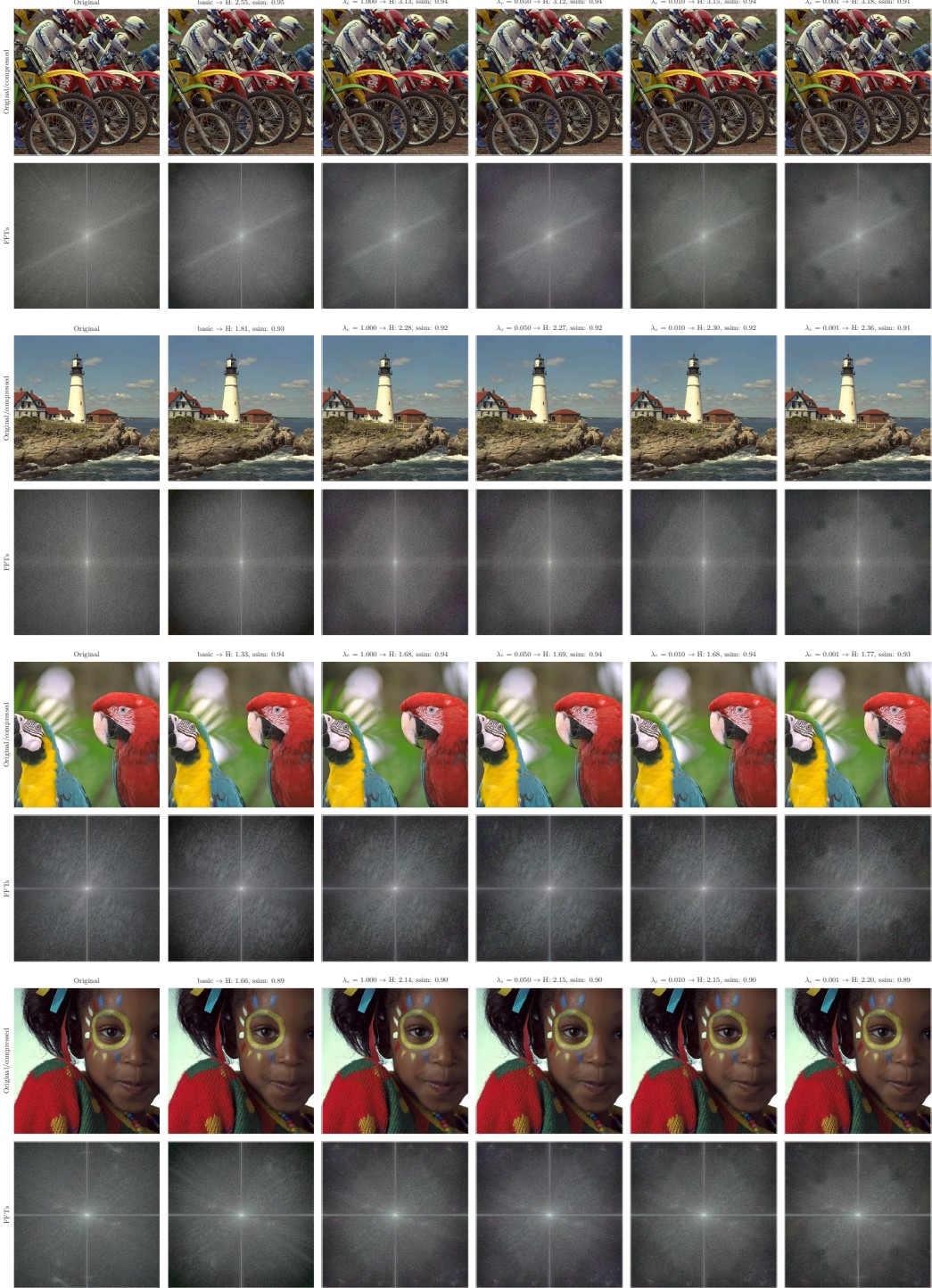

Figure B.5: Changes in the image compressed with various versions of the medium-quality DCN codec: (1st column) sample image from the *kodak* dataset; (2nd) pre-trained DCN model; (3rd-6th) fine-tuned models with decreasing $\lambda_c$. Images are ordered by SSIM fidelity loss of $\lambda_c = 1.0$ w.r.t. the pre-trained model.

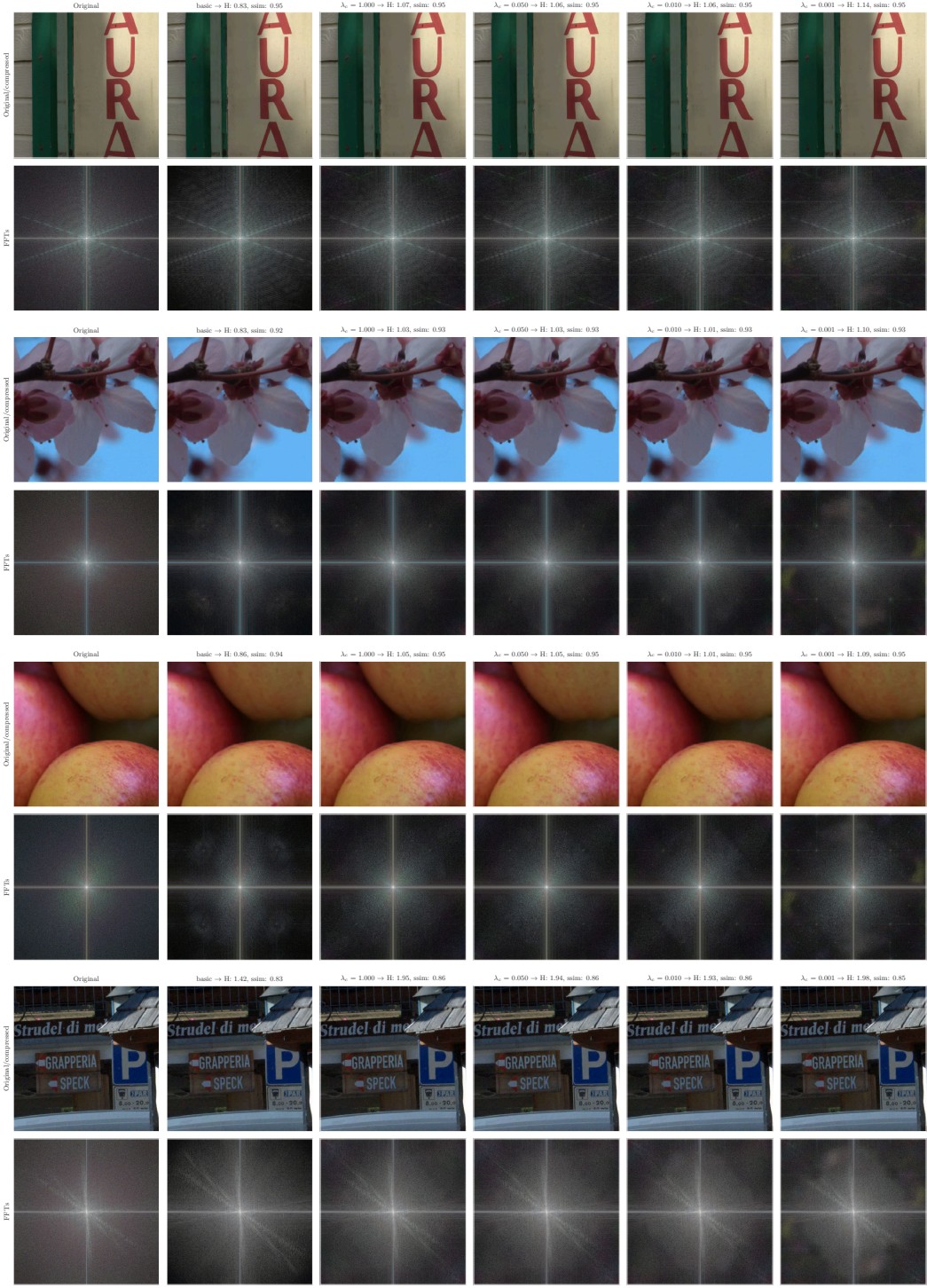

Figure B.6: Changes in the image compressed with various versions of the medium-quality DCN codec: (1st column) sample image from the *raw* dataset; (2nd) pre-trained DCN model; (3rd-6th) fine-tuned models with decreasing $\lambda_c$. Images are ordered by SSIM fidelity loss of $\lambda_c = 1.0$ w.r.t. the pre-trained model.

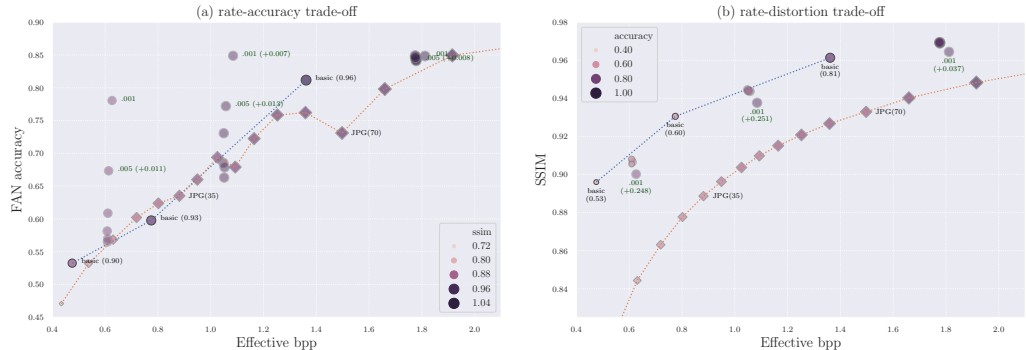

Figure C.1: Visualization of the rate-distortion-accuracy trade-off on the *clic* dataset after fine-tuning on mixed natural images.

## C  FINE-TUNING ON MIXED NATURAL IMAGES

As discussed in Section 5, we ran additional experiments by skipping photo acquisition and fine-tuning directly on mixed natural images (MNI) - a subset of the original DCN training set (2,500 images). Images in this dataset tend to have more details and depict objects at a coarser scale, since they were all down-sampled to $256 \times 256$ px (from various original sizes). This required adjusting manipulation strength to maintain visual similarity between photo variations. In particular, we used weaker sharpening, Gaussian filtering with a smaller standard deviation (0.5), down&up-sampling via 75% of the image size (instead of 50%), Gaussian noise with standard deviation 0.012, and JPEG quality level 90. We fine-tuned for 600 epochs.

We summarize the obtained results in Fig C.1 using images from the *clic* test set. In this experiment, the gap in manipulation detection accuracy between JPEG and baseline DCN has disappeared, except for remaining sudden drops at selected JPEG quality levels (corresponding to the manipulation quality factor 90). We still observe significant improvement for fine-tuned DCN models, but here it tends to saturate around 86%, which might explain negligible improvement of the high-quality 64-C model. By inspecting confusion matrices, we see most of the confusion between *native*, *sharpen* and *awgn* classes where the differences are extremely subtle.

The fine-tuned DCN models remain close to the baseline rate-distortion behavior. Interestingly, except for the weakest regularization ($\lambda_c = 0.001$), all fine-tuned models seem to be equivalent (w.r.t. the trade-off). We did not observe any obvious artifacts, even in the most aggressive models. The only image with deteriorated SSIM is the *alejandro-escamilla-6* image from *clic*, which was consistently the most affected outlier in nearly all fine tuned models for both NCO and MNI (1st row in Fig. C.2). In some replications it actually managed to improve, e.g., for the shown cases with $\lambda_c = 0.005$ and 0.001. However, we don't see any major differences between these variations.

Visualization of frequency attenuation patterns (Fig. C.3) shows analogous behavior, but the changes are more subtle on MNI. We included additional difference plots w.r.t. baseline and weakly fine-tuned models, where the changes are better visible. On NCO, due to less intrinsic high-frequency content, the behavior is still very clear (cf. bottom part of Fig. C.3).

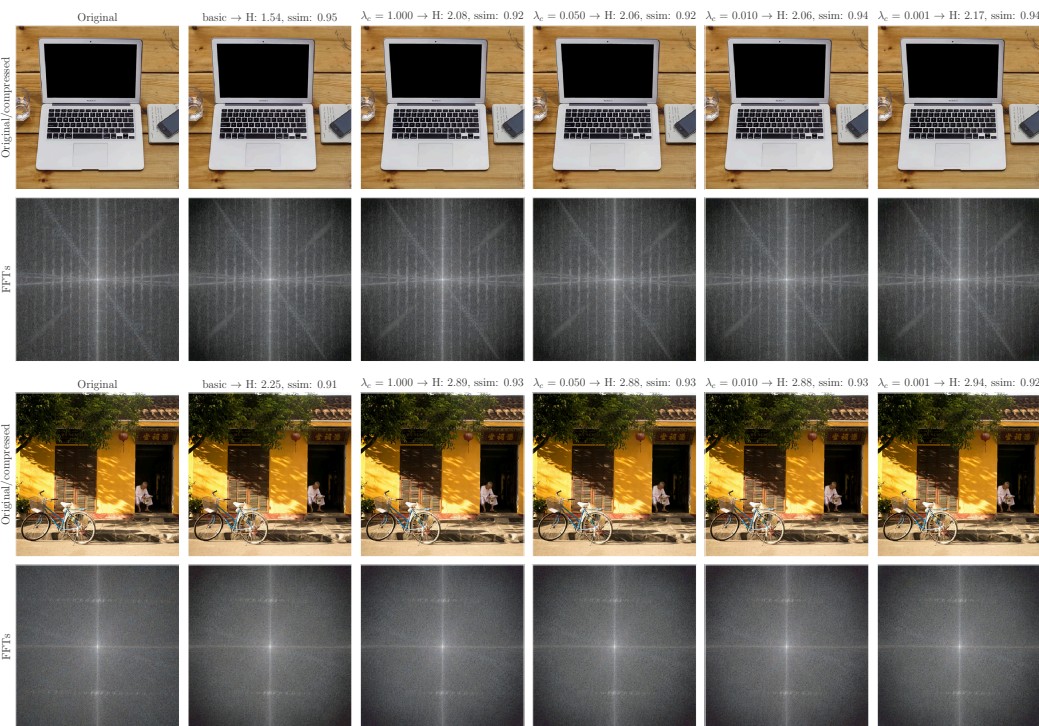

Figure C.2: Changes in images compressed with various versions of the medium-quality DCN codec fine-tuned on MNI: (1st column) sample image from the *clic* dataset; (2nd) pre-trained DCN model; (3rd-6th) fine-tuned models with decreasing $\lambda_c$. The top image corresponds to the most consistent outlier with the worst SSIM degradation.

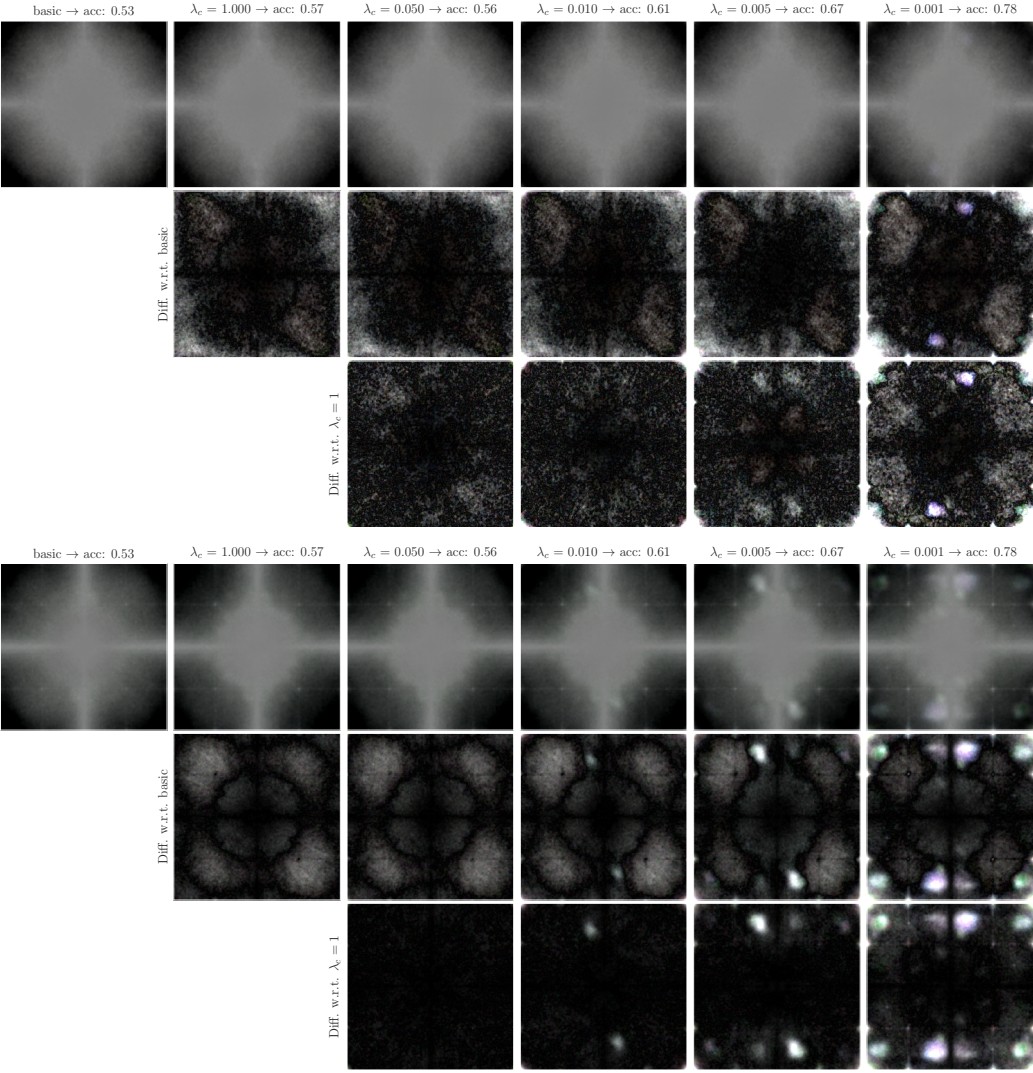

Figure C.3: Visualization of frequency attenuation/amplification patterns for DCN codecs fine-tuned on MNI: (top) low-quality codec tested on *clic* images; (bottom) the same codec tested on *raw* images. Difference plots show changes w.r.t. the baseline and weakly fine-tuned models.

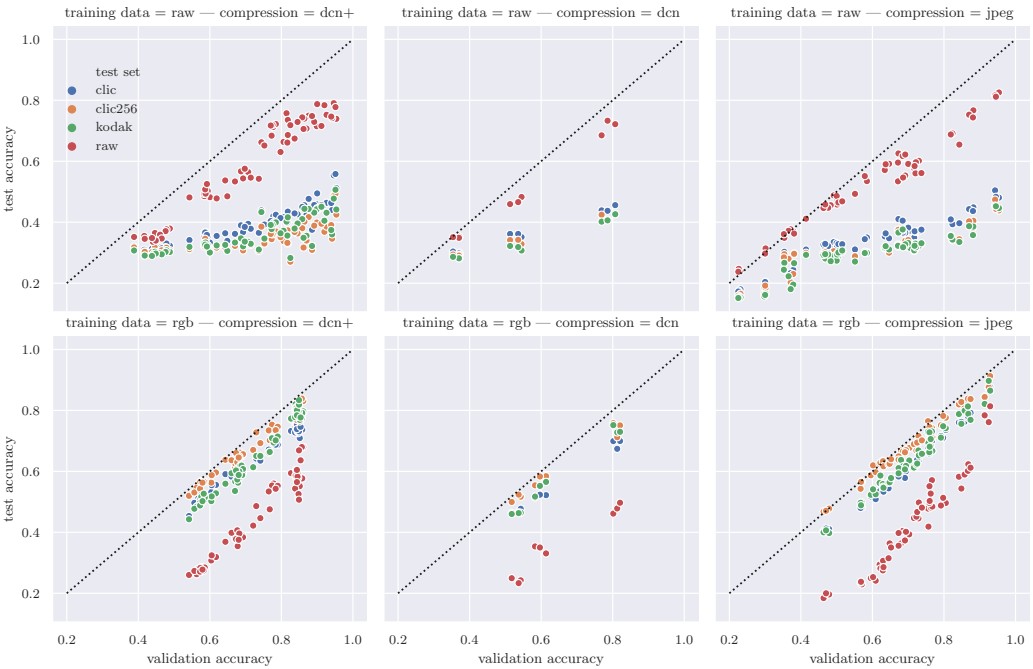

Figure D.1: Transferability of the trained FAN models to different data distributions: (top) models trained on native camera output can generalize to *raw* test images from 4 different cameras; (bottom) models trained on diverse images generalize well to different except native camera output.

## D    TRANSFERABILITY OF THE FAN

In this study, we considered two broad classes of images: *native camera output* (NCO) and *mixed natural images* (MNI) which exhibit significant differences in pixel distribution. For DCN pre-training, we relied on a large MNI dataset with images down-sampled to $256 \times 256$ px (Section 3.4). Fine-tuning was then performed on either NCO from a single camera (Nikon D90; Section 4) or a smaller sub-set of the original training MNI (2,500 images; Appendix C). Finally, we considered three test sets: *raw* with NCO from 4 different cameras; *clic* and *kodak* with MNI.

We observed that the FAN models tend to have limited generalization capabilities to images with a different pixel distribution. We ran additional experiments to quantify this phenomenon, and show the obtained results in Fig. D.1 where we compare test vs validation accuracy for various test sets (we also included a version of the *clic* set resized to $256 \times 256$ px). In the top row, we show results for models trained on NCO from a single camera. We can see imperfect, but reasonable generalization to the output of various cameras (red markers). Once the data distribution changes, the performance drops significantly. Analogously, FAN models trained on MNI generalize well to other MNI datasets (*clic* and *kodak*), but fail on NCO. We see an additional bias towards images down-sampled to the same resolution as the training data (compare *clic* vs *clic-256* images), but here the difference is small and consistent - between 5.2 - 6% based on a linear fit to the data.

