# OpenReview forum: "Quantifying the Cost of Reliable Photo Authentication via High-Performance Learned Lossy Representations"
_ICLR.cc/2020/Conference — Accept (Poster)_

### Official Review · AnonReviewer2 · 2019-10-23
**Official Blind Review #772**

**Rating:** 6

**Review:**


Summary of the paper
- This work proposes a new deep-learning-based method to replace the lossy compression techniques of images., jpg.
- The work investigates the role of codec and shows that the proposed complex photo dissemination channels optimizes the codec related traits on images.
- The method achieved much better performance in compressing images compared to practically used JPEG (QF-20)

I think the paper is well written and the experiment seems to support the author's argument. Unfortunately, this field is not overlapped to my research field, and it is hard for me to judge this paper.

**Experience Assessment:**

I do not know much about this area.

**Review Assessment: Checking Correctness Of Derivations And Theory:**

I assessed the sensibility of the derivations and theory.

**Review Assessment: Checking Correctness Of Experiments:**

I assessed the sensibility of the experiments.

**Review Assessment: Thoroughness In Paper Reading:**

I read the paper at least twice and used my best judgement in assessing the paper.

---

> ### Author Response · Authors · 2019-11-14
> **Response to Reviewer #2**
>
> Thank you for your comments.

---

### Official Review · AnonReviewer1 · 2019-10-24
**Official Blind Review #1**

**Rating:** 6

**Review:**

This paper presents a learned image compression method that is able to be robust under a variety of tasks. The results aren't state of the art in terms of rate-distortion performance, but this paper has a very good analysis of the results, and has produced a very fast codec. In that sense, this is a very interesting paper that may lead to other fast methods (the other fast method they compared the runtime against - WaveOne never published a complete description).

This paper should be likely accepted, but the authors should town down the claims a bit. The results presented do NOT show that this method is better than the best hand engineered approach, despite what they claim. Even compared to BPG, which is NOT state of the art, the results are a mixed bag.

We would like to point out to the authors that the VVC codec has shown much stronger performance than BPG, and similarly the AV1 codec has surpassed the performance of BPG. Moreover, even Pik has also surpassed the performance of BPG, so just showing stronger performance than BPG is not grounds to make the claim that this method is superior to hand engineered approaches.

Moreover, as I stated earlier, this method is not even better all the time, therefore weakening the claim.

On the positives:
- the paper fully describes the architecture, unlike WaveOne
- the runtime numbers are impressive (as far as I know, there is no faster published method)
- the authors consider applications other than compression performance (such as classification performance in forensic analysis)

On the negatives, which I highly suggest that the authors fix if this paper is to be taken seriously by the community:
- please be sure to explain that SSIM is computed in <RGB | grayscale>
- please be more explicit about which loss is used during training for distortion (i.e., "we use MSE for the training loss, but stop training when SSIM converges")
- please provide PSNR numbers for the method; and ideally MS-SSIM (in decibels) instead of PSNR
- please add other neural compression methods to the graphs
- please clarify that you create a file and decode a file for each image used to create the graphs (very important topic), as opposed to using the estimated file size
- tone down the claims w.r.t. beating classical codecs

**Experience Assessment:**

I have published in this field for several years.

**Review Assessment: Checking Correctness Of Derivations And Theory:**

I assessed the sensibility of the derivations and theory.

**Review Assessment: Checking Correctness Of Experiments:**

I assessed the sensibility of the experiments.

**Review Assessment: Thoroughness In Paper Reading:**

I read the paper thoroughly.

---

> ### Author Response · Authors · 2019-11-14
> **Response to Reviewer #1**
>
> Thank you for your detailed comments and for pointing out more recent hand-crafted codecs.
>
> We apologize for the confusion. We do not claim to get better results than state-of-the-art hand-crafted solutions. In fact, in terms of the rate-distortion performance, our codec is slightly worse, although very close to BPG. This is what we meant by saying “... is competitive with best hand-engineered codecs”. We have rephrased all related statements in the abstract and the introduction to avoid the confusion and better reflect the actual results.
>
> Regarding the remaining remarks:
>
> # please be sure to explain that SSIM is computed in <RGB | grayscale>
> The SSIM was computed as the average over RGB channels. We used the default implementation in the scikit-image package. We have extended the manuscript to explain how image quality scores are calculated (Section 3.5 / Rate-distortion Trade-off).
>
> # please be more explicit about which loss is used during training for distortion (i.e., "we use MSE for the training loss, but stop training when SSIM converges")
> We repeated again in Section 3.4 that MSE was used as the training loss explicitly optimized using Adam.
>
> # please provide PSNR numbers for the method; and ideally MS-SSIM (in decibels) instead of PSNR
> We extended the results to include MS-SSIM in decibels. The new results are included in the appendix (Figures A.5 and A.6).
>
> # please add other neural compression methods to the graphs
> We apologize but due to limited time for rebuttal, we were not able to include more neural compression methods in the comparison. We hope to include more methods upon publication of the codec on github.
>
> # please clarify that you create a file and decode a file for each image used to create the graphs (very important topic), as opposed to using the estimated file size
> Yes, we fully encoded an image into a bit-stream and subsequently decoded and reconstructed the images. The bit-stream structure is already explained in Section 3.3. To address your doubt, we emphasized that this is the case in Section 3.5.
>
> # tone down the claims w.r.t. beating classical codecs
> We rephrased all statements to clarify the confusion and better reflect the actual results.

---

### Official Review · AnonReviewer4 · 2019-11-06
**Official Blind Review #4**

**Rating:** 6

**Review:**

The paper describes a pipeline for image compression which allows to reliably detect specific manipulation patterns in compressed images.  The results show that it is possible to learn image compression that performs similarly to a modern image compression algorithm while in the same time is optimized to reveal specific kinds of manipulations. The authors build upon (Korus & Memon, 2019), but use a learnable codec instead of differentiable JPEG.

The idea to regularize entropy of the latent representation of images is interesting. A method to train a well-performing image compression system which can also follow additional constraints (such as ability to reveal certain manipulations) is very valuable for practice. Unfortunately, there are already available trainable compression methods and the authors do not compare to these methods. However, in my opinion to detect manipulation in the image one should prove that visual content in some area of the image was significantly changed with respect to some original, while in the other parts of the image it was not changed. Otherwise it becomes impossible to distinguish in-camera filtering and secondary postprocessing. Basically, the authors present a method to detect whether a very particular configuration of some basic image processing filters (Gaussian blur, median filter, resampling)  was applied to the image. Therefore the particular problem formulation looks very artificial.

With regards to the experiments in the paper, I was somewhat lost. Compare the Fig. 5 and the Fig. 8. In the Fig. 8, we see a big set of possible system configurations having different manipulation detection accuracy, image quality and compression performance.  In the Fig. 5, we see a compression efficiency-image quality dependency. However, it remains unclear how do the systems represented at these two graphs relate to each other, or, in the other words, what is he mapping between points of these graphs. Next, poor performance of JPEG manipulation detection by the proposed  network does not prove that JPEG manipulation cannot be detected, it just shows that the proposed architecture does not perform well in this problem. A comparative study which relates a new system to a current state of the art is required to claim that a proposed approach is better. Finally, SSIM is not a standard way to compute image quality. MS-SSIM and PSNR are also popular, and a user study is usually recommended to claim that some method generates images of better visual quality.

Summarizing, the authors do not provide a new best-performing image compression algorithm, and neither solve a problem of image manipulation detection, but show that it is possible to learn an image compression system with some additional constraints. I believe it is an interesting contribution, and I hope the authors can improve presentation of the experiments.

**Experience Assessment:**

I do not know much about this area.

**Review Assessment: Checking Correctness Of Derivations And Theory:**

I assessed the sensibility of the derivations and theory.

**Review Assessment: Checking Correctness Of Experiments:**

I assessed the sensibility of the experiments.

**Review Assessment: Thoroughness In Paper Reading:**

I made a quick assessment of this paper.

---

> ### Author Response · Authors · 2019-11-14
> **Response to Reviewer #4**
>
> Thank you for detailed comments.
>
> Our setup involves a forensic analysis network which learns to distinguish basic image manipulations. This corresponds to a well-established foundational test scenario, which can later be built upon to deliver practical manipulation detection algorithms. Once the model learns to distinguish different image processing paths, it will respond differently to content coming from different sources. For example, if an object is inserted or removed from a photo, that region is likely to solicit a different response than the rest of the image. An anomaly detection scheme then then be used to precisely detect and pin-point the manipulation. For a state-of-the-art system built upon this principle, we can refer readers to a recent work from CVPR [1].
>
> Regarding, Fig. 5 and Fig. 8, although there is some conceptual overlap, each figure is self-sufficient and should be viewed independently. Fig. 5 shows the standard rate-distortion trade-off for the baseline DCN model as well as standard hand-crafted codecs. Fig. 8 aims to show a more complex trade-off between rate, distortion, and forensic analysis accuracy. The main goal is to show the impact of optimization for forensic analysis at different levels of its importance. Fig. 5 can be compared with Fig. A6 which uses the same protocol and shows the same codecs using the same colors.
>
> We agree that in principle we cannot prove that more reliable detection for JPEG image is not possible. However, the forensic analysis network that we used was developed for and tested on JPEG images [2]. Due to their prevalence, forensic analysis of JPEG images is well investigated. Hence, we can assume that the accuracy we observe is a good proxy for the upper bound on the achievable classification performance. We did not perform any compression-specific tweaks and used exactly the same model in all cases.
>
> Thank you for the suggestion. We have included MS-SSIM results in the current manuscript (Figures A.5 and A.6). We agree that performing a user study would be valuable. We leave this evaluation for future work.
>
> References (all of them were already referenced in the manuscript):
> [1] Wu, Yue, Wael AbdAlmageed, and Premkumar Natarajan. "ManTra-Net: Manipulation Tracing Network for Detection and Localization of Image Forgeries With Anomalous Features." In Proceedings of the IEEE Conference on Computer Vision and Pattern Recognition, pp. 9543-9552. 2019.
> [2] Bayar, Belhassen, and Matthew C. Stamm. "Constrained convolutional neural networks: A new approach towards general purpose image manipulation detection." IEEE Transactions on Information Forensics and Security 13, no. 11 (2018): 2691-2706.

---

### Decision · Program_Chairs · 2019-12-19

**Decision:**

Accept (Poster)

**Comment:**

The paper introduces a new image compression approach that preserves the patterns indicating image manipulation. The reviewers appreciate the idea and the method. Please take into account the suggestions of Reviewer1, when preparing the final version.